# Prostaglandin E2 Receptor 4 (EP4) Affects Trophoblast Functions via Activating the cAMP-PKA-pCREB Signaling Pathway at the Maternal-Fetal Interface in Unexplained Recurrent Miscarriage

**DOI:** 10.3390/ijms22179134

**Published:** 2021-08-24

**Authors:** Lin Peng, Anca Chelariu-Raicu, Yao Ye, Zhi Ma, Huixia Yang, Hellen Ishikawa-Ankerhold, Martina Rahmeh, Sven Mahner, Udo Jeschke, Viktoria von Schönfeldt

**Affiliations:** 1Department of Obstetrics and Gynecology, University Hospital, Ludwig-Maximilians-University, Marchioninistr. 15, 81377 Munich, Germany; Lin.Peng@med.uni-muenchen.de (L.P.); Anca.Chelariu-Raicu@med.uni-muenchen.de (A.C.-R.); Zhi.Ma@med.uni-muenchen.de (Z.M.); Huixia.Yang@med.uni-muenchen.de (H.Y.); Martina.Rahmeh@med.uni-muenchen.de (M.R.); sven.mahner@med.uni-muenchen.de (S.M.); 2Chongqing Key Laboratory of Human Embryo Engineering, Chongqing Reproductive and Genetics Institute, Chongqing Health Center for Women and Children, No. 64 Jin Tang Street, Yu Zhong District, Chongqing 400013, China; 3Department of Reproductive Medicine Center, Zhongshan Hospital, Fudan University, Kongjiang Rd. 1665, Shanghai 200092, China; yeyaoeryida@163.com; 4Medizinische Klinik und Poliklinik I, Klinikum der Universität München, Ludwig-Maximilians-University, 81377 Munich, Germany; Hellen.Ishikawa-Ankerhold@med.uni-muenchen.de; 5Department of Gynecology and Obstetrics, University Hospital Augsburg, Stenglinstr. 2, 86156 Augsburg, Germany; 6Center of Gynecological Endocrinology and Reproductive Medicine, Department of Gynecology and Obstetrics, Ludwig-Maximilians University of Munich, Marchioninistr. 15, 81377 Munich, Germany; viktoria.schoenfeldt@med.uni-muenchen.de

**Keywords:** unexplained recurrent miscarriage (uRM), prostaglandin E2 receptor 4 (EP4), extravillous trophoblast cells (EVTs), phosphorylating CREB (pCREB)

## Abstract

Implantation consists of a complex process based on coordinated crosstalk between the endometrium and trophoblast. Furthermore, it is known that the microenvironment of this fetal–maternal interface plays an important role in the development of extravillous trophoblast cells. This is mainly due to the fact that tissues mediate embryonic signaling biologicals, among other molecules, prostaglandins. Prostaglandins influence tissue through several cell processes including differentiation, proliferation, and promotion of maternal immune tolerance. The aim of this study is to investigate the potential pathological mechanism of the prostaglandin E2 receptor 4 (EP4) in modulating extravillous trophoblast cells (EVTs) in unexplained recurrent marriage (uRM). Our results indicated that the expression of EP4 in EVTs was decreased in women experiencing uRM. Furthermore, silencing of EP4 showed an inhibition of the proliferation and induced apoptosis in vitro. In addition, our results demonstrated reductions in β- human chorionic gonadotropin (hCG), progesterone, and interleukin (IL)-6, which is likely a result from the activation of the cyclic adenosine monophosphate (cAMP)- cAMP-dependent protein kinase A (PKA)-phosphorylating CREB (pCREB) pathway. Our data might provide insight into the mechanisms of EP4 linked to trophoblast function. These findings help build a more comprehensive understanding of the effects of EP4 on the trophoblast at the fetal–maternal interface in the first trimester of pregnancy.

## 1. Introduction

The European Society of Human Reproduction and Embryology (ESHRE) guidelines define the concept of recurrent miscarriage (RM) as two or more consecutive failed clinical pregnancies confirmed by histopathology or ultrasound before the first 20 weeks of gestation [1]. The classification of RM etiology consists of the following factors: inherited/genetic factors, endocrine, immunological, antiphospholipid syndrome (APS) and environmental factors [1]. RM represents the most challenging and complex area of reproductive research. However, the reasons for approximately 50% of RM cases remain idiopathic, known as unexplained uRM [1].

Trophoblast cells represent the primary cells which play an essential role in implantation and human placental formation, and can compromise embryonic growth and development [2]. Considering the complex situation of the placenta in fetal–maternal communication, it is easy to understand its diverse function at different stages [3]. Within the first weeks of pregnancy, the human placenta generates trophoblasts with diverse biological functions including attachment of the conceptus to the uterine wall, establishment of early nutrition and adaption of the maternal uterine vasculature [2,4]. Trophoblast derived from the trophectoderm (cytotrophoblasts) are progenitors and differentiate into extravillous trophoblasts (EVTs) and syncytiotrophoblasts (STBs), from which multiple functions develop [5]. Current research sustains that disfunction of the extravillous trophoblasts (EVTs) might lead to early and late uRM [6].

Prostaglandin E2 (PGE2) activates four plasma membrane G-protein-coupled EP receptors (EP1-EP4) which subsequently lead to activation of several signaling pathways, three nuclear receptors, and peroxisome-proliferator-activated receptors (PPARα, PPARβ/δ, PPARγ) which stimulate gene transcription in target tissues [7,8]. In regard to the role of PGE2 in reproductive medicine, there are different studies which highlight its role in ovulation, embryonic development, and early implantation failure [9,10,11]. The trophoblast and decidual cells synthesize PGE2 within the first trimester of pregnancy [12,13].

Despite the impressive amount of research investigating the role of PGE2 in the female reproductive system, there are still intriguing questions which have not been answered. Our group conducted some primary research to examine the level of PGE2 receptors (EP1, EP2, EP3, and EP4) within the maternal–fetal interface in uRM patients and normal pregnancy. Our previous studies demonstrated that EP3 expression was increased in STBs, which led to upregulation of the inflammatory microenvironment, extracellular matrix remodeling, and hormone production within the fetal–maternal interface of uRM patients [14]. Moreover, it was found that EP2 and EP4 expression was much lower at both syncytium and decidual level in samples obtained from patients who experienced a uRM [15]. In addition, the same work demonstrated that EP2 inhibition significantly impacted the regulation of inflammatory cytokines, proliferation, and secretion of hormone production in HTR-8/SVneo. Furthermore, other research indicated PPARγ decreased within the trophoblasts, and PPARγ is involved in M2 polarization in decidual macrophages in uRM, thus suggesting that downregulation of PPARγ modulated the microenvironment at the maternal–fetal interface in recurrent miscarriage [16]. Our group’s previous work found that the expression of EP4 was decreased in the syncytiotrophoblasts (STBs) within the first trimester of pregnancy in the uRM group compared to healthy controls. Nonetheless, its expression and potential function in regulating EVTs remains relevant for future studies, and the its molecular pathological mechanism in trophoblast function is still unknown.

The main aim of our study is to identify the molecular mechanism involved in EP4 signaling and its impact on trophoblast phenotype. Additionally, we will examine the potential effects of EP4 modulation of trophoblast cell functions like secretion, invasion, apoptosis, proliferation, and viability. Finally, we will analyze the expression of the expression of EP4 in the EVTs of the decidua in uRM and healthy pregnancy. We summarize the novelty of this study as follows: (1) as far as we know, this is the first study to investigate the staining of EP4 in the maternal–fetal interface in uRM and healthy pregnancy; (2) we proposed an effective way to identify the functions of EP4 in trophoblast cells in vitro; (3) we found the molecular mechanism of EP4 which modulates the functions of trophoblasts and established the role of effective gene markers for precision treatment for uRM.

## 2. Results

### 2.1. Demographic and Clinical Presentation of the Patients

The demographic and clinical data of 38 patients enrolled in our study are illustrated in Table 1. The patients were divided into two groups: uRM and normal pregnancies. Both maternal (37.76 ± 4.88 vs. 35.78 ± 5.88, *p* = 0.41) and gestational age (37.76 ± 4.88 vs. 35.78 ± 5.88, *p* = 0.66) showed no significant differences between the two groups. Similarly, gravidity (3.42 ± 1.90 vs. 3.11 ± 1.08, *p* = 0.78) and parity had no significant differences (0.94 ± 0.94 vs. 1.63 ± 1.12, *p* = 0.06).

### 2.2. EP4 Is Downregulated in the Syncytium and EVTs in First Trimester Human Placentas with uRM

We first assessed PGE2 receptor (EP1, EP2, EP3, and EP4) expression across our two groups of patients via immunohistochemistry [14,15]. The staining of the EP4 was primarily distributed through the cytoplasm of the chorionic villous tissue and decidual cells. Interestingly, our analysis showed a significant decrease in EP4 in the syncytium and decidua of the uRM group compared to the healthy control group.

To further describe these findings, double immunofluorescence staining for human leukocyte antigen G (HLA-G) and EP4 was performed. The analysis showed that EP4 was co-expressed with HLA-G, mainly located in the cytoplasm of EVTs. In addition, intense staining for EP4 was observed in invasive EVTs in the HC group’s maternal decidua compared with that of the uRM group (Figure 1B,E). Subsequently, the percentage of HLA-G+ EP4+/HLA-G+ cells was calculated with nine slides from the uRM group and healthy controls, respectively. The rate of doubled stained cells significantly declined in the uRM group than the HC group (80.11 ± 5.21% vs. 73.53% ± 2.97%, *p* = 0.014, Figure 1G). These results suggest that the downregulation of EP4 expression in EVTs might affect the invasion of EVTs into a maternal decidual layer in patients with uRM. Furthermore, this decrease might affect trophoblast functions such as proliferation, apoptosis, invasion, and secretion as well.

### 2.3. Phosphorylation CREB (pCREB) Is Involved in the Pathogenesis of uRM

The Genecards website (https://www.genecards.org/cgi-bin/carddisp.pl?gene=PTGER4#transcripts, accessed on 1 July 2021) was used to predict a transcription factor that could bind with the EP4 promoter. Our search identified several binding sites of the PTGER4 gene promoter: AP-2α, C/EBPα, CREB, PPAR-γ1, and PPAR-γ2. Among those, we selected PPAR-γ, CREB, and pCREB and performed immunohistochemical analysis on samples obtained from first trimester human placentas.

We observed expression of pCREB at the nuclear level both in the syncytium and decidual cells. More specifically, significantly decreased pCREB expression was detected in the syncytium in uRM samples (IRS 6.705 ± 2.65 vs. 4.85 ± 2.07, *p* = 0.0278, Figure 2A,B,E). Similarly, pCREB staining was significantly lower in the decidua of uRM samples (IRS 6.411 ± 2.76 vs. 4.523 ± 2.129, *p* = 0.0194, Figure 2C,D,F). Our group’s previous work found PPARγ present in the cell’s nucleus and cytoplasm and significantly lower trophoblast expression within the uRM group [16]. In contrast, at the decidual level, a significant difference of PPARγ expression was not observed (IRS 9.24 ± 0.93 vs. 8.63 ± 1.02; *p* = 0.434). There was no significant difference of CREB staining in the syncytium and decidua between the uRM group and the control group (IRS 8 ± 2.36 vs. 7.75 ± 2.51, *p* = 0.634, Figure 2G,H,K; IRS 8.07 ± 1.75 vs. 7.63 ± 1.82; *p* = 0.396, Figure 2I,G,K, respectively). Furthermore, the Spearman correlation analysis of EP4 and pCREB according to IRS scores in 38 patients indicated that pCREB in syncytium and decidual cells was significantly positively correlated with EP4 expression (r = 0;708, *p* = 0.003, Figure 2M; r = 0.593, *p* = 0.004, Figure 2N, respectively). There was no statistically significant correlation between PPARγ and EP4 expression in the syncytium and decidua (r = 0.273; *p* = 0.429; r = 0.362; *p* = 0.291, respectively), nor for CREB and EP4 (r = −0.19; *p* = 0.723; r = −0.41; *p* = 0.1788, respectively).

Triple immunofluorescence staining was applied to co-localize EP4 and pCREB expression in decidual EVTs in the first trimester placentas. EP4 was present in the cytoplasm (P, E), while pCREB was positive at both the nuclear and cytoplasm levels (Q, V) in EVTs (O, T) in the decidua of the first trimester placenta in healthy controls and uRM. Both EP4 and pCREB were co-expressed with HLA-G, especially in the healthy group compared to uRM patients (S, X). Together, our results showed that EP4 is downregulated in the EVT of uRM patients, and its expression is positively correlated with pCREB.

### 2.4. EP4 Downregulation Inhibits Proliferation and Induces Cell Apoptosis

We further investigated the effect of EP4 knockdown on the viability and proliferation of trophoblast cells in vitro. First, we assessed EP4 expression levels in three trophoblast cell lines, HTR-8/SVneo, JEG-3, and BeWo, by Western blot. EP4 protein expression was higher in HTR-8 and JEG-3 than BeWo (Figure 3A). We subsequently transfected both HTR-8/SVneo and JEG-3 cells with a specific small interfering RNA (siEP4). The efficiency of EP4 siRNA was confirmed by qRT-PCR which showed that the EP4 mRNA level was decreased by 71.7% in HTR-8/SVneo cells with EP4si RNA3 and by 73.4% in JEG-3 cells with EP4si RNA1 compared to the negative control, respectively (Figure 3B,C).

Next, we performed a viability assay using MTT that indicated that EP4 siRNA3 significant inhibited the viability rate at different time points: after 24 h (0.795 ± 0.004 vs. 0.851 ± 0.004, *p* = 0.011), after 48 h (1.211 ± 0.04 vs. 1.321 ± 0.015, *p* = 0.042) and after 72 h (1.270 ± 0.026 vs. 1.531 ± 0.049, *p* = 0.009) compared with the negative control group in HTR-8/SVneo cells (Figure 3D). There was a significant decrease in the viability curve of EP4 siRNA 1 at 48 h (1.304 ± 0.051 vs. 1.448 ± 0.034, *p* = 0.049) and 72 h (1.501 ± 0.037 vs. 1.798 ± 0.044, *p* = 0.019) compared with the negative group in JEG-3 cells (Figure 3E).

Similarly, we observed a strong effect on proliferation in HTR-8/SVneo cells after incubation for 48 h (1.152 ± 0.119 vs. 0.955 ± 0.05, *p* = 0.023; 1.152 ± 0.119 vs. 0.944 ± 0.048, *p* = 0.058, Figure 3F) and decreased proliferation of JEG-3 cells after incubation at 72 h according to the BrdU assay (1.142 ± 0.076 vs. 0.971 ± 0.037, *p* = 0.011; 1.142 ± 0.076 vs. 0.969 ± 0.081, *p* = 0.028, Figure 3G). These results demonstrated that EP4 potently promotes trophoblast cell proliferation.

Finally, to investigate whether EP4 participates in trophoblast cell apoptosis, we performed cell death detection ELISA assays, which quantitatively determine apoptosis through detection of cytoplasmic nucleosomes. The cells were transfected with siRNA for 24 h and incubated with 8-Bromo-cAMP (PKA agonist) for 8 h. EP4 knockdown increased apoptosis in HTR-8/SVneo (6.82 ± 0.426 vs. 4.15 ± 0.252, *p* = 0.029, Figure 3H) and JEG-3 cells (3.62 ± 0.19 vs. 2.53 ± 0.120, *p* = 0.028, Figure 3I) compared to the negative control group after 48 h incubation. The promoting effect was reversed when adding 8-Bromo-cAMP to HTR-8/SVneo (4.83 ± 0.329 vs. 6.82 ± 0.426, *p* = 0.029, Figure 3H), although the result was not demonstrated in JEG-3 cells (3.09 ± 0.29 vs. 3.62 ± 0.19, *p* = 0.143, Figure 3I).

### 2.5. EP4 Effects Secretion of Hormones and Cytokines

Human chorionic gonadotropin and progesterone are two essential hormones in early pregnancy secreted from trophoblasts. We measured the secretion levels of these two crucial hormones after culture with different concentration of PGE2, TCS 2510 (a selective EP4 agonist), and L-161,982 (a selective EP4 antagonist) in supernatants of HTR-8/SVneo and JEG-3 cells.

Our results showed that the level of β-hCG was increased after 36 h incubation with 1μM PGE2 compared to the blank control (4.38 ± 0·07 vs. 3.68 ± 0·33 mIU/mL, *p* = 0.028, Figure 4A) in HTR-8/SVneo cells. The levels of β-hCG were raised compared to the concentration. After 36 h incubation with 1 μM and 10 μM of TCS 2510, the production of β-hCG was elevated compared to the blank control (4.35 ± 0.19 vs. 3.68 ± 0·33 mIU/mL, *p* = 0.032, 4.41 ± 0.15 vs. 3.68 ± 0·33 mIU/mL, *p* = 0.023, Figure 4A) in HTR-8/SVneo cells. However, L-161,982 did not alter β-hCG expression in HTR-8/SVneo cells after 24 h and 36 h incubation. The level of β-hCG was also increased after 36 h incubation with 10 μM PGE2 compared to the blank control (5.07 ± 0.13 vs. 4.39 ± 0.448 mIU/mL, *p* = 0.027, Figure 4B) in JEG-3 cells. However, after incubation with 0.1, 1,10 uM of TCS 2510 did not change the concentration of β-hCG (*p* = 0.837, Figure 4B) in JEG-3 cells. The expression of β-hCG was only significant suppressed incubation with 0.1 μM of L-161,982 for 36 h compared with the blank control (3.66 ± 0.221 vs. 4.39 ± 0.448 mIU/mL, *p* = 0.036, Figure 4B) in JEG-3 cells.

Progesterone was not detected in the supernatants of PGE2, TCS 2510, and L-161,982 after 24 h and 36 h treated groups of HTR-8/SVneo cells. Progesterone expression was only increased after 24 h incubation with 10 μM PGE2 compared with the blank control (36.23 ± 1.11 vs. 31.7 ± 1.89 ng/mL, *p* = 0.037, Figure 4C). TCS 2510 raised the production of progesterone in a concentration-dependent manner. Compared with the blank control, 0.1 and 10 μM TCS 2510 significantly increased progesterone production (36.86 ± 1.08 vs. 31.7 ± 1.89 ng/mL, *p* = 0.039, 37 ± 1.82 vs. 31.7 ± 1.89 ng/mL, *p* = 0.035, Figure 4C) in JEG-3 cells. After incubation with 0.1, 1, 10 μM of L-161,982 for 24 h and 36 h, there was no significant effect on progesterone levels compared to controls in JEG-3 cells.

The levels of β-hCG attenuated significantly in the EP4 siRNA group compared to those in the control siRNA group in both HTR-8/SVneo (3.89 ± 0.378 vs. 3.05 ± 0.367 mIU/mL, *p* = 0.028, Figure 4D) and JEG-3 cells (3.69 ± 0.356 vs. 3.05 ± 0.11 mIU/mL, *p* = 0.028, Figure 4D) after incubation for 24 h. Similarly, after incubation with EP4 siRNA, progesterone was not detected in the supernatants of HTR-8/SVneo cells. The progesterone level was also lower in the EP4 siRNA group than in the control siRNA group in JEG-3 cells (31.75 ± 1.82 vs. 28.18± 0.54 ng/mL, *p* = 0.026, Figure 4E) after incubation for 48 h in JEG-3 cells. The inhibiting effect was reversed when 8-Bromo-cAMP was added (31.38 ± 0.621 vs. 35.5 ± 0.78 ng/mL, *p* = 0.028, Figure 4E).

We then examined whether the EP4 receptor affects the secretion of cytokines in vitro. IL-6, IL-8, TNF-α, IL-1β, and PAI-1 were analyzed in the supernatants of HTR-8/SVneo and JEG-3 cells which were transfected with EP4 siRNA. IL-6 secretion was significantly reduced in HTR-8/SVneo and JEG-3 cells treated with EP4 siRNA after incubation for 48 h (3.69 ± 0.356 vs. 3.05 ± 0.110 mIU/mL, *p* = 0.028, Figure 4F). In contrast, the secretion of other cytokines was not significantly changed after incubation for 48 h.

### 2.6. EP4 Modulates the Axis of cAMP-PKA-pCREB

We found that knockdown of EP4 inhibited the production of β-hCG in the supernatants of both HTR-8/SVneo and JEG-3 cells. Phosphorylated CREB and ERK1/2 are known as the downstream transcription factors of the cAMP and PKA pathways, mediating the transcription of β-hCG and progesterone.

We next analyzed whether EP4 expression affects cAMP, PKA, pCREB, and ERK1/2 which subsequently would participate in the underlying mechanisms of EP4 downregulation in uRM. Therefore, we detected the content of cAMP in cell lysis via ELISA. Our aim was to determine whether PGE2 affects cAMP production through EP4 signaling in HTR-8/SVneo and JEG-3 cells. PGE2 stimulated cAMP in a concentration-dependent manner in HTR-8/SVneo and JEG-3 cells (Figure 5A,C). Incubation with 10 μM PGE2 for 24 h promoted cAMP expression levels compared with the blank control group (4.956 ± 0.2 pg/mL vs. 3.488 ± 0.338 pg/mL, *p* = 0.029, Figure 5A) whereas 0.1, 1 μM PGE2 had no effect on cAMP production in HTR-8/SVneo cells. The levels of cAMP increased significantly in the 1 uM TCS 2510 group compared to those in the untreated group (5.742 ± 0.169 pg/mL vs. 3.488 ± 0.338 pg/mL, *p* = 0.027, Figure 5B). When 1 uM TCS 2510 was added to the medium before treatment with PGE2, the cAMP level was significantly increased compared with untreated control group (4.956 ± 0.2 pg/mL vs. 3.488 ± 0.338 pg/mL, *p* = 0.008, Figure 5B). When 0.1μM L-161,982 was added to 10 μM PGE2, the content of cAMP significantly decreased compared with the 10 μM PGE2 group in HTR-8/SVneo (7.502 ± 0.270 pg/mL vs. 3.348 ± 0.305 pg/mL, *p* = 0.027, Figure 5B). The level of cAMP could be significantly enhanced with 1 uM TCS 2510 (3.874 ± 0.209 pg/mL vs. 2.471 ± 0.418 pg/mL, *p* = 0.028, Figure 5D), to 4.237 ± 0.448 pg/mL with PGE2 + TCS 2510 (*p* = 0.007, Figure 5D) compared with the control after 36 h incubation in JEG-3 cells. In contrast, incubation with PGE2 + TCS 2510 for 24 h significantly promoted the content of cAMP compared with the control group, whereas PGE2 + L-161,982 did not significantly decrease cAMP production compared with the PGE2 group in JEG-3 cells.

The Western blot results revealed that EP4 knockdown suppressed PKA and pCREB (phosphorylation s133) in both HTR-8/SVneo and JEG-3 cells (Figure 5E). EP4 expression was decreased by 16.4% and 26.5% through EP4 siRNA compared to the control siRNA (each *p* < 0.05, Figure 5F). PKA expression was downregulated by 25% and 50% after EP4 siRNA compared to the negative control in HTR-8/SVneo and JEG-3 cells (each *p* < 0.05, Figure 5G). The expression of p-CREB was inhibited by 41.8% and 59.7% induced by EP4 downregulation compared to the vehicle group in HTR-8/SVneo and JEG-3 cells, respectively (each *p* < 0.05, Figure 5I). In contrast, no alteration of CREB, p-ERK1/2, and ERK1/2 expression was detected through EP4 siRNA in both HTR-8/SVneo and JEG-3 cells.

## 3. Discussion

The main finding of our study is that the expression of EP4 is downregulated in the EVTs of uRM groups and the expression of EP4 has a positive correlation with p-CREB. Interestingly, EP4 demonstrated effects on proliferation and promoted the apoptosis of trophoblast cells in vitro. Finally, downregulation of EP4 attenuated the secretion of β-hCG, progesterone, and IL-6, and activates cAMP-PKA-pCREB signaling pathway.

In our project, we explored the molecular mechanisms behind the regulation of the PGE2 receptor EP4 in trophoblasts in first trimester placentas. Liu et al. suggested that PGE2 modulates biochemical and morphological placental trophoblast differentiation during implantation and placentation [17]. Previous research demonstrated that placental hypoxia activated cAMP production, proliferation, and invasion by mediating cAMP-PKA-pCREB signaling in trophoblast cells, predominantly through EP1 and EP4 [18,19]. Our findings are consistent with previous studies that showed that knockdown of EP4 decreased the proliferation and viability of HTR-8/SVneo and JEG-3 cells in vitro by inhibiting cAMP levels. However, Biondi et al. reported that PGE2 suppressed proliferation and migration through EP2 and EP4 by increasing intracellular cAMP in HTR-8/SVneo cells. We consider that the actual role of PGE2 either stimulates or inhibits trophoblast cell proliferation depending on the concentration of PGE2 and EP receptor subtypes activated in the first trimester of EVTs. In the adenylate cyclase pathway, increased cAMP levels resulted in PKA activation and a transcriptional factor that binds to CREB transactivates the transcription of specific primary response genes that initiate cell proliferation [20,21]. PGE2 has been shown to increase colon cancer cell proliferation and motility by activating the PI3K/Akt pathway by EP4 receptor activation [22].

We found that EP4 downregulation promotes apoptosis, and PKA agonists can reverse this effect. Consistent with these data, previous studies reported that EP4 receptor blockade hindered proliferation and induced apoptosis in cervical cancer cells via the EP4-PKA-CREB pathway. Moreover, reduced PKA induced apoptosis through Bcl-2/Bax-induced apoptosis [23] and activation of the caspase-8/9/11 pathway and pyroptosis [24,25]. The PKA inhibitor blocked PGE2 through EP2- and EP4-induced apoptosis, indicating that the PKA pathway is mainly responsible for PGE2-mediated inhibition of apoptosis [26].

At the initial stages in the first trimester of pregnancy, invasive extravillous trophoblasts (iEVTs) secrete β-hCG, which stimulates trophoblast invasion, angiogenesis, and placenta immune tolerance [27]. hCG might represent a serum marker of implantation and early trophoblast invasion. Progesterone production by EVTs is crucial to fine-tune vascular remodeling and escape immune attack in the maternal decidua [28]. In primary human trophoblasts, human trophoblast derived choriocarcinoma cell line BeWo, demonstrated that increasing intracellular cAMP levels activates PKA and exchanges protein directly activated by cAMP (EPAC), both of which induce the secretion of hCG and progesterone, resulting in trophoblast syncytialization and cell fusion [29,30,31]. cAMP-dependent protein kinase A (PKA) is associated with activation of the transcription factor cAMP response element-binding protein (CREB) [32]. An essential step for CREB activation and dimerization is phosphorylation of the serine residue at position 133. It recruits the adaptor proteins CBP and p300, which act as co-activates for transcription of αHCG in villous trophoblasts [33].

Furthermore, PKA can also inhibit AMP-activated protein kinase (AMPK) activity, which is responsible for the inactivation of hormone-sensitive lipase, thereby reducing the ability of LH or PGE2 to provide cholesterol for progesterone synthesis [34]. Waclawik et al. reported that estradiol-17β and progesterone increase the production of PGE2 in response to luteinizing hormone (LH) in endometrial stromal cells [35]. These findings suggest that PGE2 may be involved in a positive feedback loop during the progesterone synthesis in cells. Moreover, studies demonstrated that PGE2 improved endometrial receptivity via EP2 and EP4, acts through autocrine or paracrine pathways in vivo, and is regulated by progesterone secretion through the cAMP/PKA pathway [36,37,38]. β-hCG promotes angiogenesis in uterine vasculature, promotes the fusion of cytotrophoblast cells and differentiation to make syncytiotrophoblast cells, and causes the blockage of any immune or macrophage action by the mother to foreign invading placental cells [39]. Generally, β-hCG and progesterone also promoted maternal immune tolerance to control EVT functions and maintain embryonic–maternal communication. According to the above results, we suggest similarity and synergetic effects among PGE2 and β-hCG and progesterone pathways in HTR-8/SVneo and JEG-3 cells, and they trigger the process via activating adenylate cyclase to increase cAMP concentrations. PGE2, through its nuclear receptor PPARγ, effects trophoblast differentiation to activate downstream targets genes, and this regulation process is mainly mediated through chorion-specific transcription factor-1 (GCM-1) and increased expression of β-hCG [40,41].

It is reported that PGE2 induces cytokines and chemokine expression for trophoblast apposition and adhesion to the decidua for implantation [42,43]. IL-6 is a cytokine known to promote the differentiation of Th2 cells and subsequent suppression of Th1 cells [44]. The Th2-derived cytokines, IL-6, induce hCG release from trophoblasts, and hCG stimulates progesterone production from the corpus luteum in pregnancy [45].

Decreased expression of IL-1β and IL-6 mRNA expression has been detected in RM patient endometrium compared to healthy controls [46,47]. Furthermore, elevated production of Th1 cytokines (interferon-γ, IL-2, IL-12, TNF-β) and low levels of Th2 cytokines (IL-6) are found in samples obtained from RM women as well [48].

There is evidence that PGE2 induces IL-6 production mediated by intracellular cAMP and PKA-pCREB in orbital fibroblasts [49]. Additionally, Hershko et al. reported that the transcriptional regulation of the IL-6 gene is complex, and CREB is involved [50]. IL-6 stimulates primary trophoblast invasion and migration, utilizing the JAK/STAT and AKT signaling pathway [51,52]. The study illustrated trophoblasts secreted IL-6 to facilitate M2 polarization of macrophages by binding to the IL-6 receptor (IL-6R) on macrophages through the STAT3 signaling pathway. In that way, the activated macrophages enhance the migration and invasion of trophoblast cells in a feedback manner at the maternal–fetal interface in normal pregnancy [53]. Our data agree with these studies, which also demonstrated that the decline of EP4 decreased secretion of IL-6 and activated the cAMP-PKA-pCREB signaling pathway in HTR-8/SVneo and JGE-3 cells in vitro.

There are still some limitations in our manuscript. The main limitations of cell culture cancer cell lines are phenotypic and molecular mechanisms research. We did not use primary trophoblast cells, but choriocarcinoma tumor cells. The lack of an animal model which is closer to the microenvironment of the maternal–fetal interface is also a shortage of this manuscript. How PGE2 exerts its effects seems highly dependent on its concentration levels and which G protein receptors are activated in trophoblast cells. Previous studies from our group demonstrated that EP2 regulated the proliferation, hormone production and secretion of cytokines in trophoblast cells [15]. EP3 signaling plays a vital role in the regulation of the inflammatory microenvironment, hormone production and extracellular matrix remodeling in the maternal–fetal interface of uRM patients [14]. We deem that more in-depth research is necessary to obtain a more accurate and comprehensive understanding of the effects of PGE2 on trophoblasts.

## 4. Materials and Methods

### 4.1. Ethics Approval and Sample Tissues

The Medical Faculty’s registered study got recognition from the ethics board under the Ludwig Maximilian University of Munich (Approval Number: 337-06, 29 December 2006). Clinical sample data collected was utilized for statistical purposes once informed consent was obtained. Tissue samples were taken from 19 participating patients with a history of two or more successive miscarriages due to unexplained factors (uRM group) and 19 healthy patients who had a lawful pregnancy termination (healthy control group) who had surgery in Munich, Germany. Exclusion criteria for the uRM group included infectious diseases, uterine anomalies, endocrinological dysfunctions, hyperprolactinemia, hyperandrogenemia, thyroidal dysfunctions, autoimmunologic disorders, deficiencies in coagulation factors, as well as fetal and parental chromosomal disorders.

### 4.2. Immunohistochemistry

Our team has previously described immunohistochemical (IHC) staining [15]. Baked placentas were deparaffinized in xylol for 20 min, then incubated in methanol with H_2_O_2_ for another 20 min to block endogenous peroxidase reaction before being rehydrated in an ethanol gradient. A pressure cooker saturated with sodium citrate (pH = 6.0) was used for storing the slides. Elimination of unspecified indissoluble elements of the main antibodies was carried out, with all slides going through a blocking treatment residue (Reagent 1, Zytochem-Plus HRP-Polymer-Kit (mouse/rabbit), Berlin, Germany) about twenty minutes after being washed in PBS. The main antibody was stored overnight at 4 °C on each slide. Table 1 lists all antibodies utilized. Following the manufacturer’s guidelines, the reagent from a detection kit, the Avidin–Biotin Complex (ABC) kit (Vector Lab; Burlingame; CA; USA), was used to measure reactivity after a 1 h immersion. Immunostaining was visualized with the substrate and the chromogen-3, 3′-diaminobenzidine (DAB; Dako; Santa Clara; CA; USA) for chromogen for 1 min and hematoxylin for nuclear staining for 2 min. Distilled water was used to terminate the reaction. A semi-quantitative approach was used for the immunoreactive score (IRS) procedure; all slides were read for interpretation of saturation and allocation using a Leitz (Wetzlar, Germany) microscope. The IRS computation was calculated through addition of the percentage of positively stained cells (zero: zero pigmentation; one: mild pigmentation; two: intermediate pigmentation; three: ranges higher than 50 percent but less than 80 percent pigmentation; four: higher than eighty percent pigmentation by the saturation of cell pigmentation (0: none; 1: mild pigmentation; 2: medium pigmentation; 3: high pigmentation)).

### 4.3. Immunofluorescence Staining

The placenta specimens were filtered in xylol for twenty minutes before rinsing in 100%, 70%, 50%, and finally distilled water. A sodium citrate neutralizer (pH = 6.0) was present in the pressurized pot. Slides were washed in PBS and probed for 15 min through an agent (Ultra V Block, Lab Vision, Fremont, CA, USA). The positive control was incubated for 4 h at 4 °C overnight. Table 2 lists all antibodies utilized. After rinsing, the slides were stored in darkness for 30 min with secondary antibodies at ambient temperature. The nucleus was stained with DAPI after the slides had been thoroughly rinsed. An Axioskop fluorescent photomicroscope (Zeiss; Oberkochen, Germany) was utilized for double immunofluorescence investigation of the extravillous trophoblasts of the placenta, and pictures were captured with the Axiocam camera system (Zeiss CF20DXC). Similar paraffin-embedded slides were used together with a triple for immunofluorescence staining of EP4 and pCREB co-expression in the extravillous trophoblasts of the placenta. Microscopic confocally laser scanned photos were captured using a Zeiss LSM 880 connected to an Airyscan quality sample imaging with analysis using ZEN blue software.

### 4.4. Cell Culture

HTR-8/SVneo (ATCC CRL-3271, Manassas, VA, USA), an immortalized human first trimester placental cell line, was derived by transfection of first trimester human trophoblasts with a gene encoding simian virus 40 large T antigen. Choriocarcinoma-derived placenta trophoblast cell lines JEG-3 (ATCC HTB-36, Manassas, VA, USA) and BeWo cells (ATCC CCL-98, Manassas, VA, USA). The three cells were grown on RPMI1640 medium + Gluta MAX (Gibco; McKinley Place NE; MN; USA) combined with 10% fetal bovine serum (FBS, Gibco; McKinley Place NE; MN; USA). The cells were refined in a humid incubator at 37 °C under 5% carbon dioxide. Replacement of the cultural mediums occurred between 2 and 3 days. Cells were planted in each well of a 96-well, with saturation of cell viability and multiplication studies. ELISA testing, hormone measurement, and wound healing assays were all performed in 24-well plates. Cells were cultivated in 6-properly saturations to aid in Western blotting and for real-time polymerase chain reaction (RT-PCR).

### 4.5. Treatment with EP4 siRNAs

The method was previously described in full [54]. Cells were cultured in 6-well or 96-well saturations in RPMI 1640 solution with 10% FBS, growth averaging 60% confluence, and EP4 and controlling siRNAs were retrieved at a concentration of 20 nM through OriGene setup (CAT: SR321501, Rehovot, HaMerkaz, Israel), transferred utilizing Lipofectamine 3000 connected at a final concentration of 50 nM with producer protocols (Invitrogen, Waltham, MA, USA). For all subsequent tests, cells were immersed in new solution with or without solamargine for up to another 24 h after being cultured, adding up to 30 h.

### 4.6. Real-Time Quantitative Extraction

Using a RNeasy Mini Kit, total RNA was extracted from cells (Qiagen, Hilden, Germany). RNA was transformed into a first-strand cDNA manufacturing kit following the producer’s protocol (Epicenter, Madison, WI, USA). With FastStart Essential DNA Probes Master and gene-specific primers (Applied Biosystems, Hs00259261 CE for EP4), the expression of EP4 mRNA was measured using RT-PCR. Next, 1 μL TaqMan^®^ Gene Expression Assay (Thermo Fisher Scientific, Waltham, MA, USA) 20×, 1μL cDNA, 8 μL H2O (DEPC-treated DI water), and 10 μL TaqMan^®^ Fast Universal PCR Master Mix 2× were combined in a 20 μL reaction mixture. Then, a 20 L TaqMan^®^ Gene Expression Assay and TaqMan^®^ Fast Universal PCR Master Mix experiment was conducted on an Optical Fast 96-well plate with an optical sticky film over it, 2 μL cDNA template and 8 μL RNase-free water were created in each probe. The temperature regimen was 95 °C for 20 s, 95 °C for 3 s, and 60 °C for 30 s, accompanied by 40 cycles of intensification. The housekeeping genes β-actin (Nr. Hs99999903 mL) were employed as reference controls for expression normalization, and the comparative CT technique was employed for computation.

### 4.7. Western Blot Analysis

The cells were immersed in a 200 L catalyzing solution containing a 1:100 dilution of protease inhibitor (Sigma-Aldrich, Merck Darmstadt, Germany) in RIPA buffer for 30 min (Sigma Aldrich, R0278, Merck, Darmstadt, Germany). The lysate was centrifuged, and the supernatant was subjected to a Bradford protein test. Protein separation occurred at 10% SDS-PAGE, with transference into nitrocellulose cells according to their molecular weights (Bio-Rad, Feldkirchen, Germany). Non-specific binding was prevented by incubating the membrane for 1 h with 4% skim milk powder before incubating for 16 h at 4 °C with the main antibodies (the antibodies utilized are listed in Table 1). The samples and secondary antibodies are stored for 1 h at ambient temperature after being washed three times for 10 min in TBS/Tween. Detection was performed with 5-bromo-4-chloro-3′-indolylphosphate/nitro-blue tetrazolium (BCIP/NBT) chromogen substrate solution (Promega, Walldorf, Germany). Western blots were scanned and quantified by densitometry, using the GelScan V6.0 1D Analysis Software (SERVA, Electrophoresis GmbH, Heidelberg, Germany).

### 4.8. Cell Viability Assays and Cell Proliferation

For 24 h, HTR-8/SVneo and JEG-3 cells were seeded in 96-well plates and transfected with EP4 siRNA or control siRNA. Cell credibility was assessed using the MTT (3-(4, 5-dimethylthiazol-2-yl) 2, 5-diphenyl tetrazolium bromide) test (Roche, Mannheim, Germany). Under cultivation conditions, the MTT-labeled reagent was applied for complete saturation of 0.5 mg/mL about 30 min. The optical density (OD) was measured using 595 nm through an Elx800 universal Microplate Reader. HTR-8/SNveo and JEG-3 microorganisms developed into 96-well sections for the proliferation test. The cells were then treated with 0.1, 1, 10 nM PGE2, TCS 2510, and L-161,982 the following day. As a blank control, DMSO was used. The manufacturer’s protocol was followed to measure cell proliferation using the 5-Bromo-2′-Deoxyuridine (BrdU) incorporation assay (11647229001, Roche). An Elx800 universal Microplate Reader was used to test color intensity of 450 nm. Each experiment was carried out three times.

### 4.9. ELISA of cAMP Assay

The intracellular cAMP levels were determined using an enzyme-linked immunosorbent test kit (R&D system, KGE012B, Minneapolis, MN, USA). The color development was halted, and the absorbate were assessed at 450 nm. The cAMP concentration was calculated using ng/mL. The cAMP measurement for each stimulation section was adjusted to the control group cAMP measurements.

### 4.10. Cell Death Detection ELISA

Using the Cell Death Determination ELISA Kit (Roche, cat# 11544675001), apoptotic cell death was collected from HTR-8/SVneo and JEG-3 cells cultivated in triplicate with four treatment groups. Using the manufacturer’s protocol, the kit quantifies cell death using a colorimetric test that measures the proportion of cytoplasmic histone-associated DNA fragments. An Elx800 universal Microplate Reader was utilized for measurement of absorbance at 460 mm.

### 4.11. β-hCG and Progesterone Measurement

Supernatants were collected from 24-well plates of HTR-8/SVneo, JEG-3, and BeWo cells that had been incubated in defined medium for 24 and 36 h, respectively, and refluxed at 10,000× *g* for 10 min for filtration of cell sections. β-hCG and progesterone were gauged using the recommended guidelines with an ADVIA Centaur XP auto analyzer (Siemens Medical Solution Diagnostics, Erlangen, Germany), described in previous studies [17].

### 4.12. Detection of Cytokines by ELISA

After 24 h of transfection with si-EP4 or si-Control, the cells centrifuged at 1000× *g* for 10 min. The concentrations of IL-6, IL-8, TNF-α, PAI-1, and IL-1 in culture solution were determined with ELISA kits (R&D Systems, Minneapolis, MN, USA) following the production protocol. The absorbance readings were measured using an Elx800 universal Microplate Reader.

### 4.13. Statistic Analysis

The terms used are the mean ± standard deviation (SD). To conduct analytical assessments, the GraphPad Prism program (version 8.0, San Diego, CA, USA) was utilized. The correlation between the two indicators was examined using Pearson’s correlation assessment.

The Student’s t-test was used for the comparison of maternal age and gestational age in uRM patients and healthy controls. A Mann–Whitney test was performed for gravidity and parity in two groups. A Mann–Whitney test was applied for evaluating the IRS scores of pCREB, CREB and PPARγ expression and the percentage of double staining cells in the placentas of the two groups. A Wilcoxon test was used for evaluation of the proliferation rate, apoptosis rate, β-hCG, progesterone, IL-6, IL-8, TNF-α, IL-1, cAMP and PAI-1 expression levels between vehicle and stimulation groups. A Wilcoxon test was also used for analyzing the band intensities of EP4, PKA, p-CREB, CREB and β-actin. At *p* < 0.05, differences were considered statistically significant (* *p* 0.05; ** *p* 0.01; *** *p* 0.005).

## 5. Conclusions

In conclusion, our study demonstrated that EP4 is decreased in EVTs in uRM patients. Interestingly, knockdown of EP4 inhibited proliferation, promoted apoptosis, and reduced the secretion of β-hCG, progesterone, and IL-6 in in vitro trophoblast cell culture models, which was associated with the cAMP-PKA-pCREB signaling pathway (Figure 6). Our future in-depth mechanistic studies will address the issue of the role of PGE2 and its specific EP receptor in immune cells (macrophages, dendritic cells, regulatory T cells, and natural killer cells) and crosstalk between trophoblasts and immune cells in the decidua during the first trimester placentas of uRM patients.

## Figures and Tables

**Figure 1 ijms-22-09134-f001:**
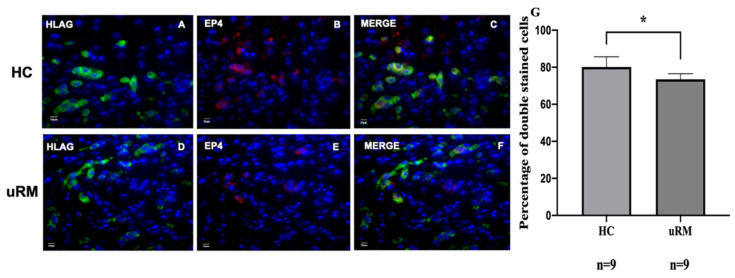
Double immunofluorescence analyses of EP4 and HLAG shown in the decidua of uRM patients and healthy controls. EP4 is co-expressed with HLAG in the cytoplasm of extravillous trophoblast cells (**A**–**F**). The percentage of double staining cells were counted in HLAG^+^ EP4^+^/HLAG^+^ cells over nine slides. The expression of EP4 in EVTs was decreased in uRM patients compared to healthy controls (**G**). HLAG is specific marker for extravillous trophoblast cells (EVTs). The picture is shown in its original magnification of 40×. * *p* < 0.05.

**Figure 2 ijms-22-09134-f002:**
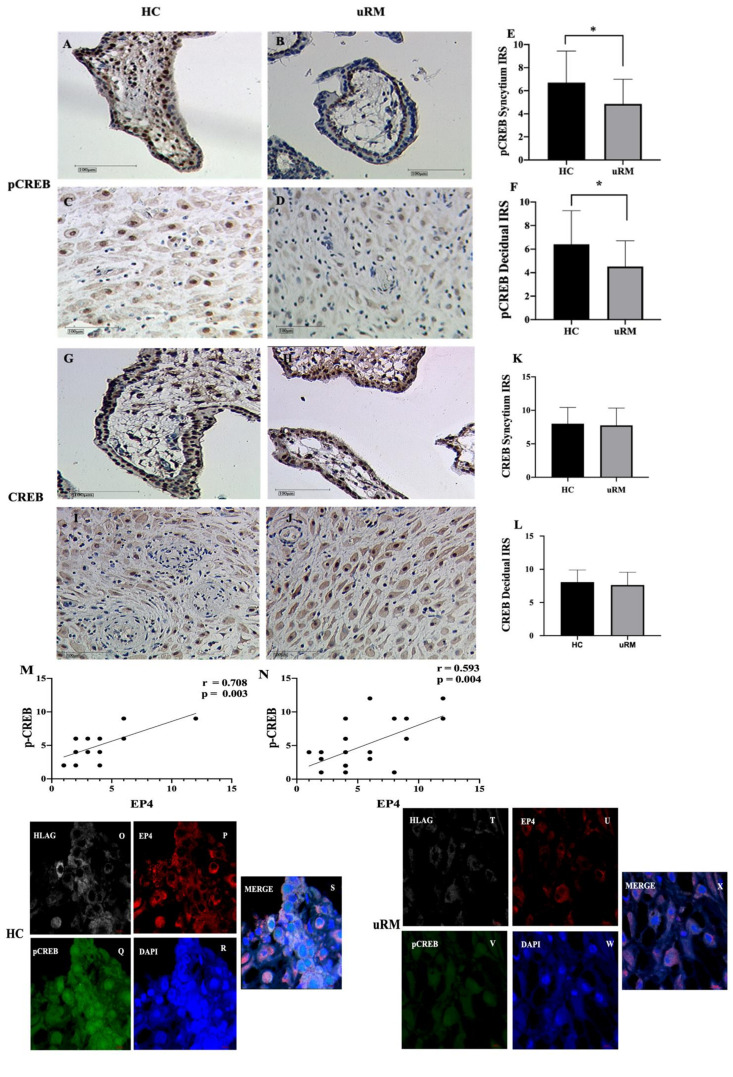
Immunohistochemical analysis of pCREB and CREB expression in placentae of uRM patients and healthy controls from the first trimester were assessed via IRS score. Expression of pCREB and CREB was identified in the nuclear of cells in the syncytium (**A**,**B**,**G**,**H**) and in decidual (**C**,**D**,**I**,**J**) of first-trimester placentas in both healthy control and uRM. The expression of pCREB is decreased in both the syncytium and decidual cells (**E**,**F**) were measured via IRS score. EP4 is significantly positively correlated with pCREB in both the syncytium and decidua (**M**,**N**). Confocal microscopy imaging showed that EP4 (red) (P, E) co-expresses and co-localizes with p-CREB (green) (**Q**,**V**) in extravillous trophoblasts (EVTs) (O,T) of uRM patients (**S**) and health controls (**X**). HLAG is specific marker for EVTs. The picture was shown in its original magnification of 40×.

**Figure 3 ijms-22-09134-f003:**
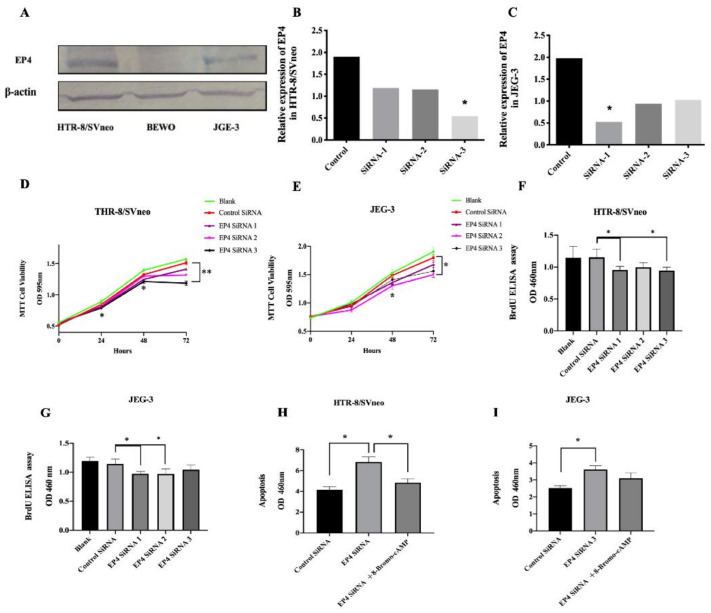
EP4 knockdown inhibits proliferation and induced apoptosis in vitro. (**A**). The expression protein EP4 is higher in HTR-8/SVneo and JEG-3 cells than BEWO. (**B**). The mRNA expression of EP4 in HTR-8/SVneo after knockdown with siRNA was detected by RT-PCR. (**C**) The mRNA mRNA of EP4 in JEG-3 after knockdown with siRNA was detected by RT-PCR. (**D**) MTT assays indicated that downregulated EP4 decreased the viability of HTR-8/SVneo compared to the negative control. (**E**) The MTT assay indicated that downregulated EP4 decreased the viability of JEG-3 compared to the negative control. (**F**) BrdU assay suggested that the proliferation of HTR-8/SVneo is inhibited by EP4 knockdown compared to the negative control. (**G**) BrdU assay suggested that the proliferation of JGE-3 is inhibited by EP4 knockdown compared to negative control. (**H**) The apoptosis test showed that the apoptosis of HTR-8/SVneo with EP4 siRNA is improved compared to the negative control and the effect was reversed by 8-Bromo-cAMP. (**I**) The apoptosis test showed that the apoptosis of JGE-3 with EP4 siRNA is reduced compared to the negative control. * *p* < 0.05.

**Figure 4 ijms-22-09134-f004:**
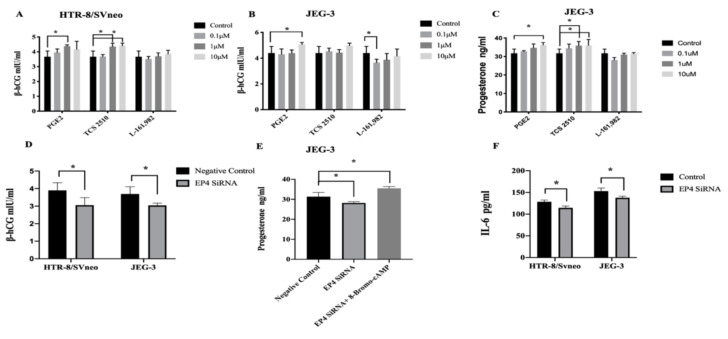
The expression levels of β-hCG, progesterone, and IL-6 in the supernatants of HTR-8/SVneo and JEG-3 cells after incubation with PGE2, TCS 2510, L-161,982 and siRNA. (**A**) The secretion of β-hCG was increased by 1 μM PGE2, 1 μM and 10 μM TCS 2510 in THR-8/SVneo. (**B**) The secretion of β-hCG was increased by 10 μM PGE2 and inhibited by 0.1 μM L-161,982 in JEG-3. (**C**) The production of progesterone in JEG-3 was increased by 10 μM PGE2, 1 μM and 10 μM TCS 2510. (**D**) β-hCG levels in the supernatants of HTR-8/SVneo and JEG-3 cells were dropped after slicing EP4 compared to negative control. (**E**) Progesterone levels were decreased in the supernatants of JEG-3 cells after slicing EP4 and EP4 added 8-Bromo-cAMP compared to negative controls. (**F**) IL-6 levels were attenuated in the supernatants of HTR-8/SVneo and JEG-3 cells after knockdown of EP4 compared to negative controls. * *p* < 0.05.

**Figure 5 ijms-22-09134-f005:**
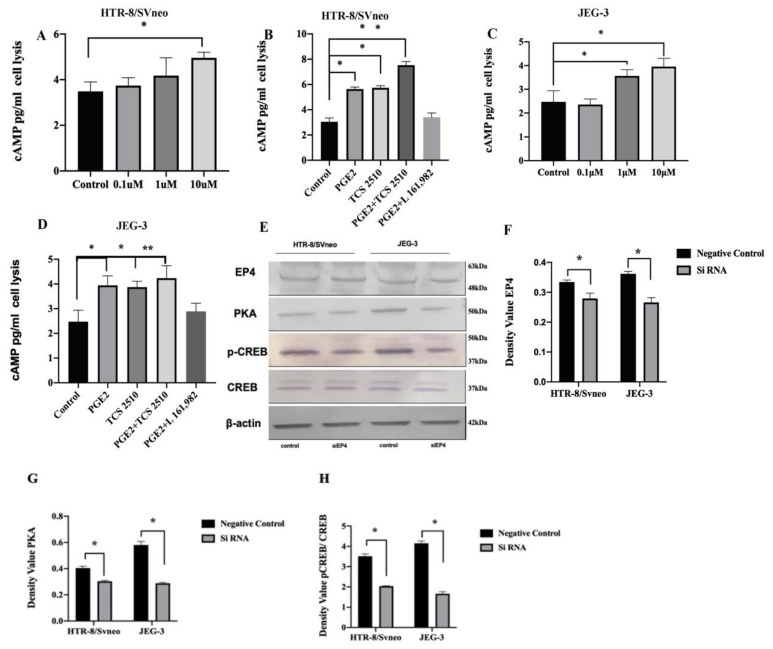
The expression levels of cAMP, EP4, PKA and p-CREB were changed by EP4 knockdown. (**A**) The secretion of cAMP was increased by 10 μM PGE2 in HTR-8/SVneo.(**B**) The secretion of cAMP was increased by PGE2, TCS 2510, PGE2 + TCS 2510 and inhibited by PG2 + L-161,982 in HTR-8/SVneo. (**C**) The production of progesterone in JEG-3 was increased by 1, 10 μM PGE2. (**D**) cAMP levels in the cell lysis of JEG-3 was increased by PGE2, TCS 2510, PGE2 + TCS 2510. (**E**) Western blot analysis showed EP4, PKA, p-CREB, CREB in HTR-8/SVneo and JEG-3 cells after silencing the EP4 gene compared with the negative control. (**F**) The density value of EP4 in HTR-8/SVneo and JEG-3 cells after knockdown of EP4 compared to negative control. (**G**) The density value of PKA in HTR-8/SVneo and JEG-3 cells after knockdown of EP4 compared to negative control. (**H**) The density value of PKA in HTR-8/SVneo and JEG-3 cells after knockdown of EP4 compared to negative control. * *p* < 0.05; ** *p* < 0.01.

**Figure 6 ijms-22-09134-f006:**
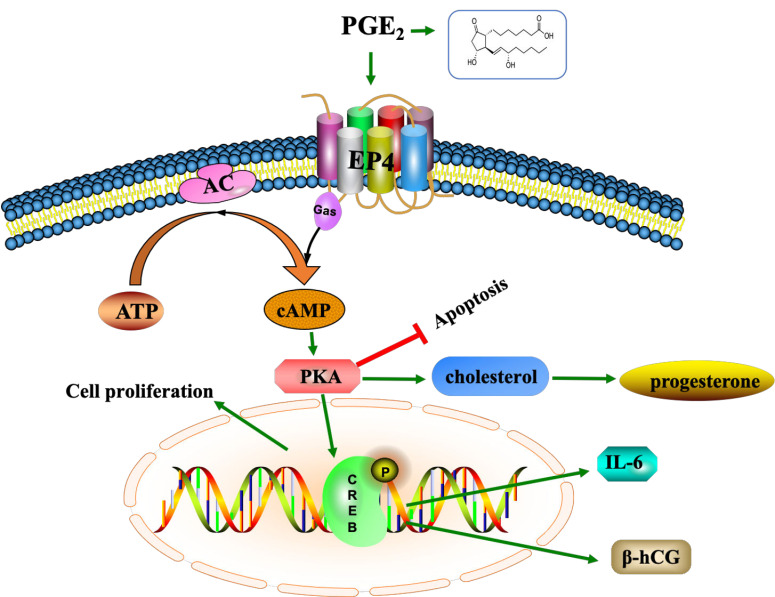
The possible role of EP4 in trophoblasts of unexplained recurrent miscarriage (uRM). Inhibiting the EP4 signaling pathway reduces the activity of the cAMP-PKA-pCREB signaling pathway, which can eventually lead to depressing the production of progesterone, β-hCG and IL-6. Downregulated of EP4 decreased proliferation and increased apoptosis in trophoblasts. These abnormal changes in trophoblasts may contribute to recurrent miscarriages.

**Table 1 ijms-22-09134-t001:** Demographic and clinical characteristics of the study population.

Characteristic	Normal Pregnancy *n* = 19	uRM *n* = 19	*p* Value
maternal age (years)	35.78 ± 5.88 (25–46)	37.76 ± 4.88 (30–44)	0.41
gestational age (weeks)	9.71 ± 1.88 (6–13)	9.09 ± 2.17 (4–12)	0.66
gravidity	3.42 ± 1.90 (1–7)	3.11 ± 1.08 (2–5)	0.78
parity	1.63 ± 1.12 (0–4)	0.94 ± 0.94 (0–3)	0.06

**Table 2 ijms-22-09134-t002:** Antibodies used in this study.

Antibody	Species	Clone	Dilution	Company (Catalog#)	Concentration (μg/mL)
EP4	rabbit	polyclonal	1:100	Abcam. ab45295	400
CREB	rabbit	monoclonal	1:500	Abcam. ab32515	166
pCREB	rabbit	monoclonal	1:1000	Abcam. ab32096	150
PPAR-γ	mouse	monoclonal	1:500	Abnova. MAB8316	1000
HLA-G	mouse	monoclonal	1:200	Novus. NBP1-43123	1000
HLA-G biotin	mouse	monoclonal	1:200	LSBio. LS-C204117	NA
Cy-2	goat	polyclonal	1:500	Dianova. 115-225-146	1500
Cy-3	goat	polyclonal	1:100	Dianova.115-167-003	1000
Cy-5	Streptavidin	NA	1:200	Dianova.016-170-084	1000
PKA	rabbit	monoclonal	1:5000	Abcam, ab32514	200
pPKA	rabbit	polyclonal	1:500	Abcam, ab226754	300
β-actin	mouse	monoclonal	1:1000	Sigma, A5441	200

NA = Not applicable.

## Data Availability

Data can be made available by the corresponding author upon reasonable request.

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
