# Peer review of "Prostaglandin E2 Receptor 4 (EP4) Affects Trophoblast Functions via Activating the cAMP-PKA-pCREB Signaling Pathway at the Maternal-Fetal Interface in Unexplained Recurrent Miscarriage"

_ijms, 2021, doi:10.3390/ijms22179134_

Round 1
Reviewer 1 Report
An interesting study; I have a number of comments.
Line 36: typo – data, not date
Line 54: human EVT don’t proliferate; please amend this sentence
Lines 51-55: The introduction lacks the required depth of information. Please remind readers about the trophectoderm surrounding the implanting embryo, the three types of trophoblast it forms – the syncytiotrophoblast (STB), cytotrophoblast (CTB) and extravillous trophoblast (EVT), and the structure of placental villi. This information is needed for context and to critically evaluate the models you have used. It would also be useful to define “successful implantation” and state that adhesion and stable attachment of the embryo to the decidua is needed prior to placental development.
Line 74-75: which trophoblasts are you referring to herre? EVT, CTB or STB? Or HTR-8s?
Line 78-79: Do you mean EP4 is decreased in the STB of women with a history of RM? You mention this is observed across all 3 trimesters, but if women have RM, they lose their pregnancy in the first 20 weeks and don’t reach the third trimester.
Line 100 – you need to define your exclusion criteria in this manuscript, even as a supplementary table, so the reader doesn’t have to access another publication to understand the design of this study.
ine 103 – “Baked placentas”? Do you mean fixed placentas? How was the tissue processed on collection, separated and fixed? What size pieces of tissue were fixed? Was everything fixed, or were random samples taken? How were random samples selected?
Line 104: do you mean xylene? Is this a typo?
Line 110: Table 1 (antibody concentrations) is missing from the manuscript. Make sure antibody concentrations are shown in ug/ml and catalogue numbers are included on this table. The table of patient demographic details should be renamed as Table 2.
Lines 114-115: It looks like your tissues are counterstained. What with? Please add the method to this section
Lines 141-144. This sentence is poorly worded and misleading. Please revise it. Also, BeWo cells are also a choriocarcinoma cell line, not an EVT cell line; please correct this. Describe HTR-8/SVneo cells correctly – they are an immortalised human first trimester EVT-like cell line.
Line 145 and beyond. Please describe the cell lines as cells, not microorganisms. Microorganisms is the wrong term to use.
Lines 155-156 “…developed in 6-well or 96-well saturations…” do you mean “cells were cultured in 6-well or 96-well plates? If so, correct all of the text accordingly.
Lines 157-159: Please provide the concentration of the siRNAs and lipofectamine used, as well as the catalogue number for the siRNAs.
PCR: Please provide the primer sequences and concentrations used.
Western blotting: After antibody binding, how were bands visualised? More methodological detail is needed.
Lines 195-196? Did you not add DMSO before measuring the absorbance?
Line 199. Check concentrations of PGE2, TCS2510 and L161 in this sentence – do you mean M? Concentration of 0.1M, 1M and 10M for all of these molecules seem far too high. What was the rationale for using these concentrations? Are they biologically relevant concentrations?
Line 221: what are HTV cells? You have not mentioned these before
Lines 234 – 241: Your choice of statistical analysis appears to be incorrect in places. Please describe what normality testing was undertaken to determine which of your datasets were parametric. Please note that any normalised data (e.g. percentages, fold change from control, relative expression) are by definition non-parametric, and should be expressed as medians and analysed by Mann Whitney, Tukey’s or Kruskal Wallis tests, as appropriate. N numbers are absent from your figure legends; please also note that for low n numbers, it is not possible to conduct an accurate normality test so data should be considered non-parametric and analysed accordingly. Please add n numbers and statistical used tests for every graph shown to all of the figure legends.
Line 256: Chorionic, not chronic
Figure 1: The images are quite small; a higher magnification would be helpful to see the dual staining better. Please also increase the magnification of the placenta IHC images in Figure 2.
Figure 2M and N: There should be a single, linear correlation line on these graphs; please update.
Lines 335-340: You should make clear that EVT do not proliferate, thus the effects on proliferation you are reporting are relevant only to villous cytotrophoblasts, not EVT.
Discussion and conclusions – please describe your findings appropriately. JEG-3 are choriocarcinoma cells and HTR-8 cells are a model of first trimester EVT. Use this language when describing your results; do not ascribe your findings directly to primary EVT as this is misleading.
Please address the limitations of the study. You used trophoblast cell lines; this work really needs to be repeated in primary cells/tissues to validate your conclusions as there are a number of publications showing how poor a representation of primary tissue various placental cell lines are. What else does PGE2 do? Can it signal through other receptors and modulate other signalling pathways e.g. EP2? Are you certain the functional effects of PGE2 are mediated via EP4?
Why have you not cited the paper by your group published in J Reprod Immunol in 2020, as the study is very similar to this one, and your findings should be discussed in the context of signalling via the EP2 receptor.
The manuscript needs to be checked by a native English speaker, as there are typos, grammatical errors and unusual and non-scientific phrasing throughout.
Author Response
Responds to the reviewer’s comments: Reviewer #1:
- Response to comment: Line 36: typo – data, not date
Response: We revised the date to data.
- Response to comment: Line 54: human EVT don’t proliferate; please amend this sentence.
Response: We have made correction according to your suggestions. Current research sustains that disfunction of extravillous trophoblast (EVTs) might lead to early and late uRM.
- Response to comment: Lines 51-55: The introduction lacks the required depth of information. Please remind readers about the trophectoderm surrounding the implanting embryo, the three types of trophoblast it forms – the syncytiotrophoblast (STB), cytotrophoblast (CTB) and extravillous trophoblast (EVT), and the structure of placental villi. This information is needed for context and to critically evaluate the models you have used. It would also be useful to define “successful implantation” and state that adhesion and stable attachment of the embryo to the decidua is needed prior to placental development.
Response: It is really true as Reviewers suggested that Trophoblast cells represent the primary cells which play essential role in implantation and human placental formation, can compromise embryonic growth and development[2]. Considering the complex situation of the placenta in fetal-maternal communication, it is easy to understanding its diverse function at different stages[3]. Within the first weeks of pregnancy the human placenta generates trophoblast with diverse biological functions including attachment of the conceptus to the uterine wall, establishment of the early nutrition and adaption of the maternal uterine vasculature[2, 4]. Trophoblast derived from trophectoderm, cytotrophoblasts are progenitors and differentiated into extravillous trophoblast (EVT) and syncytiotrophoblast (STB) with multiple functions develop[5]. Current research sustains that disfunction of extravillous trophoblast (EVTs) might lead to early and late uRM [6].
- Response to comment: Line 74-75: which trophoblasts are you referring to herre? EVT, CTB or STB? Or HTR-8s?
Response: we referred to syncytiotrophoblasts (STBs) here.
- Response to comment: Line 78-79: Do you mean EP4 is decreased in the STB of women with a history of RM? You mention this is observed across all 3 trimesters, but if women have RM, they lose their pregnancy in the first 20 weeks and don’t reach the third trimester.
Response: Yes, EP4 is decreased in the STB of women in the first trimester of placenta with 12 weeks. We are very sorry for our incorrect writing.
- Response to comment: Line 100 – you need to define your exclusion criteria in this manuscript, even as a supplementary table, so the reader doesn’t have to access another publication to understand the design of this study.
Response: We added the exclusion criteria as your suggested. Exclusion criteria for the uRM group, including infectious diseases, uterine anomalies, endocrinological dysfunctions, hyperprolactinemia, hyperandrogenemia, thyroidal dysfunctions, autoimmunologic disorders, deficiencies in coagulation factors, as well as fetal and parental chromosomal disorders.
- Response to comment: line 103 – “Baked placentas”? Do you mean fixed placentas? How was the tissue processed on collection, separated and fixed? What size pieces of tissue were fixed? Was everything fixed, or were random samples taken? How were random samples selected?
Response: Sorry for that mistake. “Paraffine embedded formalin fixed tissue was used to perform tissue slides. The slides (3mm) were deparaffinized in xylol for 20 minutes, then incubated in methanol with H2O2 for another 20 minutes to block endogenous peroxidase reaction before being rehydrated in an ethanol gradient.”
The placenta was formalin fixed immediately after curettage. The whole placenta was used, no pre-selection.
- Response to comment: Line 104: do you mean xylene? Is this a typo?
Response: Yes, it’s a typo. Line 104: the statement of “xyolo” was corrected as “xylol”.
- Response to comment: Line 110: Table 1 (antibody concentrations) is missing from the manuscript. Make sure antibody concentrations are shown in ug/ml and catalogue numbers are included on this table. The table of patient demographic details should be renamed as Table 2.
Response: We are very sorry for our incorrect writing.
Table 1 Antibodies used in this study
|
Antibody |
Species |
Clone |
Dilution |
Company (Catalog#) |
Concentration (ug/ml) |
|
EP4 |
rabit |
polyclonal |
1:100 |
Abcam. ab45295 |
400 |
|
CREB |
rabit |
monoclonal |
1:500 |
Abcam. ab32515 |
166 |
|
pCREB |
rabit |
monoclonal |
1:1000 |
Abcam. ab32096 |
150 |
|
PPAR-γ |
mouse |
monoclonal |
1:500 |
Abnova. MAB8316 |
1000 |
|
HLA-G |
mouse |
monoclonal |
1:200 |
Novus. NBP1-43123 |
1000 |
|
HLA-G biotin |
mouse |
monoclonal |
1:200 |
LSBio. LS-C204117 |
NA |
|
Cy-2 |
goat |
polyclonal |
1:500 |
Dianova. 115-225-146 |
1500 |
|
Cy-3 |
goat |
polyclonal |
1:100 |
Dianova.115-167-003 |
1000 |
|
Cy-5 |
Streptavidin |
NA |
1:200 |
Dianova.016-170-084 |
1000 |
|
PKA |
rabit |
monoclonal |
1:5000 |
Abcam, ab32514 |
200 |
|
pPKA |
rabit |
polyclonal |
1:500 |
Abcam, ab226754 |
300 |
|
β-actin |
mouse |
monoclonal |
1:1000 |
Sigma, A5441 |
200 |
NA= Not Applicable
- Response to comment: Lines 114-115: It looks like your tissues are counterstained. What with? Please add the method to this section.
Response: Immunostaining was visualized with the substrate and the chromogen-3, 3′-diaminobenzidine (DAB; Dako) for chromogen for 1 minute and hematoxylin for nuclear staining for 2 minutes.
- Response to comment: Lines 141-144. This sentence is poorly worded and misleading. Please revise it. Also, BeWo cells are also a choriocarcinoma cell line, not an EVT cell line; please correct this. Describe HTR-8/SVneo cells correctly – they are an immortalised human first trimester EVT-like cell line.
Response: HTR-8/SVneo (ATCC CRL-3271, Manassas, USA), an immortalized human first-trimester EVT cell line, is derived by transfection of first trimester human trophoblasts with gene encoding simian virus 40 large T antigen. Choriocarcinoma-derived placenta trophoblast cell lines JEG-3 (ATCC HTB-36, Manassas, USA) and BeWo cells (ATCC CCL-98, Manassas, USA).
- Response to comment: Line 145 and beyond. Please describe the cell lines as cells, not microorganisms. Microorganisms is the wrong term to use.
Response: We made the corrections as your comments.
- Response to comment: Lines 155-156 “…developed in 6-well or 96-well saturations…” do you mean “cells were cultured in 6-well or 96-well plates? If so, correct all of the text accordingly.
Response: We have rewritten this sentence according to your suggestion.
- Response to comment: Please provide the concentration of the siRNAs and lipofectamine used, as well as the catalogue number for the siRNAs. PCR: Please provide the primer sequences and concentrations used. Western blotting: After antibody binding, how were bands visualised? More methodological detail is needed.
Response: The siRNAs were transfected into cells using Lipofectamine 3000 (Invitrogen, Waltham, MA, USA) at a final concentration of 50nM and siRNAs were used at 20nM (CAT: SR321501, Rehovot, HaMerkaz, Israel). For primer sequences, primer assays or TaqMan probes for EP4 (hs00259261): forward 5 -TGTAAAACGACGGCCAGT, reverse 5 -CAGGAAACAGCTATGACC. 1 μl TaqMan® Gene Expression Assay 20 x, 1μl cDNA, 8 μl H2O (DEPC treated DI water), and 10 μl TaqMan® Fast Universal PCR Master Mix 2x are combined in a 20μl reaction mixture.Method for western blotting bands visualized: Detection was performed with 5-bromo-4-chloro-3′-indolylphosphate /nitro-blue tetrazolium (BCIP/NBT)-chromogen substrate solution (Promega). Western blots were scanned and quantified by densitometry, using the GelScan V6.0 1D Analysis Software (SERVA, Electrophoresis GmbH, Heidelberg, Germany).
- Response to comment: Lines 195-196? Did you not add DMSO before measuring the absorbance?
Response: Yes, MTT assays for EP4 siRNA and control did not add DMSO.
- Response to comment: Line 199. Check concentrations of PGE2, TCS2510 and L161 in this sentence – do you mean M? Concentration of 0.1M, 1M and 10M for all of these molecules seem far too high. What was the rationale for using these concentrations? Are they biologically relevant concentrations?
Response: We are very sorry for our incorrect writing. The “concentration of 0.1M, 1M and 10M” was amended as “concentration of 0.1nM, 1nM and 10nM”.
- Response to comment: Line 221: what are HTV cells? You have not mentioned these before
Response: We are very sorry for our wrong writing.
- Response to comment: Lines 234 – 241: Your choice of statistical analysis appears to be incorrect in places. Please describe what normality testing was undertaken to determine which of your datasets were parametric. Please note that any normalised data (e.g., percentages, fold change from control, relative expression) are by definition non-parametric, and should be expressed as medians and analysed by Mann Whitney, Tukey’s or Kruskal Wallis tests, as appropriate. N numbers are absent from your figure legends; please also note that for low n numbers, it is not possible to conduct an accurate normality test so data should be considered non-parametric and analysed accordingly. Please add n numbers and statistical used tests for every graph shown to all of the figure legends.
Response: Thanks so much for your patient explanation and professional revision on this issue. We have made related changes of the paper as your advice.
The periods are terms of mean ± Standard Deviation (SD). To do analytical assessments, the GraphPad Prism program (version 8.0, San Diego, CA, USA) was utilized. The correlation between the two indicators was examined using Pearson's correlation assessment. Student's t-test was used for the comparison of maternal age and gestational age in uRM patients and healthy controls. Mann–Whitney test was performed for the comparison of gravidity and parity in two groups. Mann–Whitney test was applied for evaluating IRS scores of pCREB, CREB and PPARγ expression and percentage of double staining cells in placentas of two groups. Wilcoxon test was used for the evaluation of proliferation rate, apoptosis rate, β-hCG, progesterone, IL-6, IL-8,TNF-α,IL-1, cAMP and PAI-1 expression levels between vehicle and stimulation groups. Wilcoxon test was also used for analyzing the band intensities of EP4, PKA, p-CREB, CREB and β-actin. At P< 0.05 differences were considered statistically significant (*P 0.05; **P 0.01; ***P 0.005).
- Response to comment: Line 256: Chorionic, not chronic
Response: The word chronic was corrected as chorionic.
- Response to comment: Figure 1: The images are quite small; a higher magnification would be helpful to see the dual staining better. Please also increase the magnification of the placenta IHC images in Figure 2.
Figure 2M and N: There should be a single, linear correlation line on these graphs; please update.
Response: We have made corresponding changes as your advice.
- Response to comment: Lines 335-340: You should make clear that EVT do not proliferate, thus the effects on proliferation you are reporting are relevant only to villous cytotrophoblasts, not EVT.
Discussion and conclusions – please describe your findings appropriately. JEG-3 are choriocarcinoma cells and HTR-8 cells are a model of first trimester EVT. Use this language when describing your results; do not ascribe your findings directly to primary EVT as this is misleading.
Response: We have corrected some inaccurate expression; thus, these changes will improve the manuscript.
- Response to comment: Please address the limitations of the study. You used trophoblast cell lines; this work really needs to be repeated in primary cells/tissues to validate your conclusions as there are a number of publications showing how poor a representation of primary tissue various placental cell lines are. What else does PGE2 do? Can it signal through other receptors and modulate other signalling pathways e.g. EP2? Are you certain the functional effects of PGE2 are mediated via EP4?
Why have you not cited the paper by your group published in J Reprod Immunol in 2020, as the study is very similar to this one, and your findings should be discussed in the context of signalling via the EP2 receptor.
Response: We added the limitations of the manuscript as well as revised it according to the comments. There are still some limitations in our manuscript. The main limitations of cell culture cancer cell lines are section of the phenotypic and molecular mechanism research. The use of unexplained recurrent miscarriage-derived primary trophoblast cells is more creditable than choriocarcinoma tumors cells. Lack of animal model which is closer to the microenvironment of maternal-fetal interface also shortage of this manuscript.
PGE2 through its’ nuclear receptor PPARγ affecting trophoblast differentiation to activate downstream targets genes, and this regulation process is mainly mediated
through chorion-specific transcription factor-1 (GCM-1) and the increased expression of β-hCG.
PGE2 exerts its effects seem highly dependent on its concentration levels and which G protein receptors are activated in trophoblast cells. Previous researches of our group demonstrated that EP2 regulated the proliferation, hormone production and secretion of cytokines in trophoblast cells. EP3 signaling plays a vital role in the regulation of the inflammatory microenvironment, hormone production and extracellular matrix remodeling in the maternal-fetal interface of uRM patients. We deem that more in-depth research is necessary to obtain more accurate and comprehensive understanding of the effects of PGE2 on trophoblasts. We are sorry to forget to cite the EP2 manuscript in the discussion.
Special thanks to you for your good comments.

Reviewer 2 Report
Whilst in my opinion the study is scientifically sound the English and presentation lets the the paper down and must be addressed before publication. Clearly the results and Discussion have been written by someone completely different to the earlier introduction and methods sections.
I have some concerns about statistics and presentation of data which I would also like the authors to address. My full comments are detailed in the attached document.

Author Response
Reviewer #2:
- Response to comment: page 1 abstract line 26 and 27 delete ‘also’ insert comma’ i.e ‘ produce, amongst other molecules, prostaglandins.
Response: We have deleted ‘also’ and added comma in the abstract.
- Response to comment: line 35 delete ‘in great detail’
Response: line 35‘in great detail’ was deleted.
- Response to comment: line 37 replace ‘dare” with data
Response: We are so sorry for your incorrect writing.
- Response to comment: [REVIEWER COMMENT re last line in abstract
‘Therefore, increased EP4 expression might be potentially therapeutic strategy for trophoblast dysfunction caused by uRM patients.’
The big question is how you would safely induce EP4 expression in endometrial tissue of a women to treat for unexplained recurrent miscarriages? I would suggest this speculative comment is removed from the abstract.]
Response: The statement “Therefore, increased EP4 expression might be potentially therapeutic strategy for trophoblast dysfunction caused by uRM patients.”were corrected as “These findings help us more and comprehensive understanding of the effects of EP4 on trophoblast at the fetal-maternal interface of first trimester of pregnancy”.
- Response to comment: page 2 line 46-47, unless the authors are alluding to somatic mutations, inherited and genetic mean the same thing. Suggest the syntax ‘inherited/genetic factors’ instead of ‘inherited and genetic’ . Also
Response: As the reviewer suggested that the inherited/genetic factors is appropriate.
- Response to comment: line 47-48 delete ‘genetic’….it makes no sense here before ‘and environmental’
Response: We deleted the word ‘genetic’.
- Response to comment: line 50 delete ‘known as’ insert ‘remain’ [this sufficiently implies the defines of the abbreviation uRM]
Response: We have made correction according to your comments, which is concise and clear.
- Response to comment: Line 60 do the authors mean within the first trimester rather than ‘three semesters’
Response: Yes, our sample is in the first trimester of pregnancy.
- Response to comment: Line 67 suggest replacing ‘Our publication explained ‘with “Our previous studies have demonstrated…”
Response: Line 67, the statement of ‘Our publication explained’ was corrected as ‘Our previous studies have demonstrated’.
- Response to comment: line 74 replace ‘another’ with ‘other
Response: Line 74, the word replaces ‘other’ replaced ‘another’.
- Response to comment: Line 76 -78 ‘, thus suggesting that downregulate of PPARγin recurrent miscarriage is preferably linked to the specific inflammatory response against the fetus[13]’. I am not at all clear as to what the authors are trying to say here. Please rewrite this, and as a separate sentence “Thus suggesting that down-regulation of PPARγin recurrent miscarriage is ??????????? linked to [WHAT] specific inflammatory response against the fetus[13]
Response: We have re-written this sentence. Furthermore, other research indicated a PPARγ decreased within the trophoblasts, and PPARγ is involved in M2 polarization in decidual macrophages in uRM, thus suggesting that downregulate of PPARγ modulated the microenvironment at the maternal-fetal interface in recurrent miscarriage.
- Response to comment: Line 81 - 82 “regulating the EVTs remains relevant for future studies, and the its molecular pathological mechanism in trophoblast function is still unknown.
Response: Line 81 – 82, ‘its’ was added.
Thank you again for your positive comments on our manuscript. Hopefully, we could have our article been considered of publication in your journal. Should there been any other corrections we could make, please feel free to contact us.

Round 2
Reviewer 1 Report
Thank you for addressing the majority of my comments, the manuscript is much improved as a result. However, I still believe that the manuscript lacks novelty, given that you have chosen not to repeat any of your findings using human placental explants or primary cytotrophoblast cultures.
Author Response
Dear Reviewer and editor,
We appreciate the careful reading and valuable suggestions. We carefully considered the comments and have revised the manuscript accordingly.
- The subchapter explaining the novelty of the study.
We summarize the novelty of this study as follows: (1) as far as we know, this is the first study to explore the staining of EP4 in the maternal-fetal interface in uRM and healthy pregnancy. (2) we proposed an effective way to identify the functions of EP4 in trophoblast cells in vitro. (3) we found the molecular mechanism of EP4 to modulate the functions of trophoblast and established its role of effective gene markers for precision treatment for uRM.
- the limitation that the study did not use primary cells in the discussion.
We did not used primary trophoblast cells but choriocarcinoma tumors cells.
Once again, thank you very much for your comments and suggestions.
Sincerely yours
Udo Jeschke
